# Deep Learning-Based Algorithm for Road Defect Detection

**DOI:** 10.3390/s25051287

**Published:** 2025-02-20

**Authors:** Shaoxiang Li, Dexiang Zhang

**Affiliations:** School of Electrical Engineering and Automation, Anhui University, Hefei 230601, China; a15398230527@163.com

**Keywords:** road defect detection, YOLOv8, GD mechanism, RepViTBlock, Wise-IoU loss function

## Abstract

With the increasing demand for road defect detection, existing deep learning methods have made significant progress in terms of accuracy and speed. However, challenges remain, such as insufficient detection precision for detection precision for road defect recognition and issues of missed or false detections in complex backgrounds. These issues reduce detection reliability and hinder real-world deployment. To address these challenges, this paper proposes an improved YOLOv8-based model, RepGD-YOLOV8W. First, it replaces the C2f module in the GD mechanism with the improved C2f module based on RepViTBlock to construct the Rep-GD module. This improvement not only maintains high detection accuracy but also significantly enhances computational efficiency. Subsequently, the Rep-GD module was used to replace the traditional neck part of the model, thereby improving multi-scale feature fusion, particularly for detecting small targets (e.g., cracks) and large targets (e.g., potholes) in complex backgrounds. Additionally, the introduction of the Wise-IoU loss function further optimized the bounding box regression task, enhancing the model’s stability and generalization. Experimental results demonstrate that the improved REPGD-YOLOV8W model achieved a 2.4% increase in mAP50 on the RDD2022 dataset. Compared with other mainstream methods, this model exhibits greater robustness and flexibility in handling road defects of various scales.

## 1. Introduction

As China’s economy takes off and develops, infrastructure development has been steadily advancing and achieving remarkable results. In particular, great achievements have been made in the field of highways. By the end of 2023, the country’s highway mileage had reached 5,436,800 km, an increase of 82,000 km from the end of the previous year [1]. The increasing mileage of highways also raises the question of how to maintain them. As highways are used more and more, both asphalt and cement pavements will gradually develop defects due to the erosion of the natural environment and the influence of human activities. Early common defects are cracks and potholes [2]. As the road ages, these defects will become more severe, posing a serious threat to the safety of pedestrians and vehicles. The timely detection and repair of road defects is crucial to ensuring road safety and prolonging pavement lifespan.

Early defect detection techniques have relied mainly on manual visual inspection [3]. Despite the simplicity and directness of this method, the accuracy of visual inspection is limited by the subjective judgement of the inspector, making it inefficient, time consuming and difficult to meet the increasing demand. With the expansion of roads and the scale of traffic, this method is increasingly unable to meet the accuracy and efficiency requirements of modern industrial inspection.

In contrast, multi-purpose inspection vehicles [4] based on integrated sensors such as GPS, cameras, laser profilers, ground-penetrating radar, etc., are more accurate and efficient. These vehicles are able to carry out inspections without interrupting normal traffic, greatly improving the convenience and accuracy of inspection work. Since the beginning of the 21st century, a number of countries have gradually introduced their own road defect detection vehicles, and Roadware has also developed a night inspection vehicle [5]. However, the high cost limits the popularity of these vehicles, especially in rural areas where the budget is limited and the relevant departments can hardly afford to use this kind of equipment, so the application of road inspection vehicles in rural areas is still limited. To overcome this problem, some scholars have tried to explore the use of 3D sensors for road defect detection, which has advantages in terms of accuracy and detail detection, but its equipment and application costs are still high and unsuitable for wider use [6].

In recent years, with the rapid progress of deep learning, target detection techniques have become increasingly mature, demonstrating remarkable prediction accuracy and detection speed. Most research results [7,8,9] indicate that, as long as the training data are sufficiently large and a well-designed algorithm is used, the performance of artificial intelligence can be nearly comparable to that of humans, making it capable of handling various detection tasks. Meanwhile, techniques such as knowledge distillation [10,11], transfer learning [12], and model resizing enable an effective balance between accuracy, speed, and computational resource consumption, leading to their widespread application [13,14,15,16] across various fields. In particular, the extensive use of convolutional neural network (CNN) [17] has significantly advanced road defect detection technology. Compared with traditional feature extraction methods, deep learning greatly enhances the accuracy and efficiency of defect detection by automatically learning image features [18].

Cao Jingang et al. [19] proposed a crack detection network (ACNet) based on the attention mechanism, using ResNet34 as the feature extraction backbone and introducing the attention feature module (AFM) to enhance the detection of multi-scale cracks. The attention-based decoding module (ADM) was designed to achieve accurate positioning of cracks. Miao Ren et al. [20] proposed a diagonal IoU loss function for optimizing the regression computation of the bounding box to enhance the detection capability. Meanwhile, the YOLOv5 model was improved by combining the Generalized and Decoupled Head modules, which achieve high- and low-level feature efficient fusion, thereby improving the detection accuracy. Hu et al. [21] proposed a YOLOM lightweight pavement disease detection method by fusing state-space models (SSM), with the aim of resolving the problem of insufficient accuracy of existing detection methods. The study designed a visual Mamba layer with multiple scanning modes and improved the rapid extraction of image features by optimizing the data normalization method to adapt to the small batch training. In addition, parallel computing units were designed to accelerate network computation and reduce algorithm training time. In Guo et al.’s study [22], MobileNetV3 was introduced to replace YOLOv5 as the base network, and the K-means clustering algorithm was used to optimize and filter the a priori frames. This method not only improves the accuracy of target detection but also effectively reduces the computational complexity of the model, making it more lightweight, while significantly improving the computational efficiency.

The above studies show that, for the challenges of defect detection in complex road scenes, the accuracy and robustness of detection can be significantly improved by improving the algorithm structure, optimizing the feature extraction method and introducing scene-adaptive enhancement strategies. These methods can not only reduce false detections and omissions but also deal more effectively with the interference of morphological diversity and complex defect background. However, road defect detection still faces the following challenges: in complex scenes, defect features are often small and easily overwhelmed by the background information, and the detection ability of existing methods for small targets is still insufficient; in the actual road detection scene, dynamic environmental factors such as light changes, rain and snow, and vehicle and pedestrian interference can significantly affect the stability and accuracy of detection; and the shapes, textures, and distributions of road diseases are complex and varied. Therefore, how are more robust detection methods to be designed? Furthermore, the shapes, textures, and distributions of road defects are highly diverse, posing challenges for designing more robust feature extraction and classification models. Based on YOLOv8n, this paper makes the following improvements to address the above problems: (1) design a new Rep-GD module; replace the original C2f module in the GD mechanism with the improved C2f module of RepViTBlock, which achieves higher feature expression capability and lower computational complexity and significantly reduces the computational cost; (2) replace the Rep-GD module with the Neck part in the YOLOv8 model, which solves the problem of information loss in recursive transmission by aggregating and distributing global information and improves the detection capability of road defects at different scales; and (3) introduce the Wise-IoU loss function to make the model converge faster and with higher accuracy.

## 2. RepGD-YOLOV8W Algorithm Construction

### 2.1. GD

In traditional target detection tasks, FPN [23] is a commonly used feature fusion structure, but it suffers from incomplete fusion of feature information in practical applications. This is because FPN can only effectively fuse the features of neighboring layers, while for the features of non-neighboring layers can only be passed indirectly by recursion, which leads to the loss of information. To address the above problem, this paper introduces the GD mechanism proposed by Gold-Yolo [24], which can effectively aggregate global information by globally fusing features at different levels and injecting them into features at different levels, thus avoiding the problem of information loss in the recursive transfer and significantly improving the ability of the model to detect road defects at different scales. The GD mechanism consists of three core modules: feature alignment module (FAM), information fusion module (IFM), and information injection module (Inject). In the gather process, the feature alignment module (FAM) first extracts and aligns features from different layers, ensuring consistency of features in the spatial dimension and providing the basis for subsequent fusion. Next, the information fusion module (IFM) integrates the aligned features to generate global information. Once the gathering process is complete, the Information injection module (Inject) distributes the fused global features to each layer and combines the local features of that layer. Through a simple attention mechanism, the global features are effectively fused with the local features, enhancing the feature expressiveness of each layer and significantly improving the model’s recognition performance.

The GD mechanism is designed for multi-scale features with two branches, low-GD and high-GD, to cope with small and large target detection requirements, respectively.

#### 2.1.1. Low-GD

The low-GD module is mainly used to fuse the shallow feature information of the model to achieve efficient interaction of shallow features and distribution of global information through the steps of feature alignment, information fusion, and information injection. The exact process is shown in Figure 1.

In the feature alignment phase, the multi-scale features [*B*2, *B*3, *B*4, *B*5] extracted from the backbone network are unified to the target size $R_{B4}=\frac{1}{4}R$ by the low-order feature alignment module (*Low_FAM*). The *Low_FAM* uses an average pooling operation to downsample the input features and unify the sizes with the following formula:(1)$$F_{align}=Low\_FAM\left(\left[B2,B3,B4,B5\right]\right).$$

This alignment not only preserves low-level information but also balances computational complexity and feature integrity, making subsequent information fusion more efficient.

In the information fusion phase, the GD mechanism uses a multi-layer reparametrized convolutional block (Re*pBlock*) to extract and process the aligned features $F_{align}$ to generate the fused features $F_{fuse}$. Subsequently, the fused features are used to generate the injected features $F_{inj\_P3}$ and $F_{inj\_P4}$ through the segmentation operation of the channel dimensions to support the operation of the injection module:(2)Ffuse=RepBlockFalign(3)$$F_{inj\_P3},F_{inj\_P4}=Split\left(F_{fuse}\right).$$

In the information injection phase, global information is injected into the corresponding local features through the attention mechanism. The injection module inputs the local feature $F_{local}$ and the global injected feature $F_{\mathrm{inj}}$ of the current layer, for the global injected feature $F_{\mathrm{inj}}$, $F_{global-embed}$ and $F_{act}$ are obtained using two independent convolution operations, and $F_{local-embed}$ is extracted using a convolution operation for the local feature x of the current layer. Since the sizes of $F_{local}$ and $F_{global}$ may be different, $F_{act}$ and $F_{global-embed}$ are resized to match $F_{\mathrm{inj}}$ using average pooling or bilinear interpolation. The final output features are further optimized with Re*pBlock*, where $F_{local}$ is equal to Bi at the low stage, so the formula is as follows:(4)$$F_{global\_act\_Pi}=resize\left(Sigmoid\right)\left(Conv_{act}\left(F_{inj\_Pi}\right)\right)$$(5)$$F_{global\_embed\_Pi}=resize\left(Conv_{global\_embed\_Pi}\left(F_{inj\_Pi}\right)\right)$$(6)$$F_{att\_fuse\_Pi}=Conv_{local\_embed\_Pi}\left(Bi\right)*F_{ing\_act\_Pi}+F_{global\_embed\_Pi}$$(7)Pi=RepBlockFatt_fuse_Pi.

#### 2.1.2. High-GD

The high-GD module is mainly used to fuse the deep-level feature information of the model to optimize the detection performance of medium and large targets. The efficient integration of high-level information is realized by extracting and distributing the global semantic information of the deep-level features. The exact process is shown in Figure 2.

In the feature alignment stage, the multi-scale features [*P*3, *P*4, *P*5] generated by the low-GD module are selected and unified by the higher-order feature alignment module (*High_FAM*) to the minimum size of $R_{P5}$. *High_FAM* uses an average pooling operation to reduce the feature size, thus reducing the subsequent computational requirements while preserving the global context information:(8)$$F_{align}=High\_FAM\left(\left[P3,P4,P5\right]\right).$$

In the information fusion stage, a transformer block is used instead of Re*pBlock* to fully fuse the global information. Subsequently, $F_{fuse}$ is downscaled by 1 × 1 convolution, and $F_{fuse}$ is divided into $F_{inj-N4}$ and $F_{inj-N5}$ along the channel dimensions by a segmentation operation.(9)$$F_{fuse}=Transformer\left(F_{align}\right)$$(10)$$F_{inj-N4},F_{inj-N5}=Split\left(Conv1\times 1\left(F_{fuse}\right)\right)$$

In the information injection stage, the high-GD module continues the injection mechanism of low-GD, which combines global features with deep local features, completes the fusion through the attention mechanism, and extracts the final output features by Re*pBlock*. In the high stage, Flocal is equal to Pi, so the formula is as follows:(11)$$F_{global\_act\_Ni}=resize\left(Sigmoid\right)\left(Conv_{act}\left(F_{inj\_Ni}\right)\right)$$(12)$$F_{global\_embed\_Ni}=resize\left(Conv_{global\_embed\_Ni}\left(F_{inj\_Ni}\right)\right)$$(13)$$F_{att\_fuse\_Ni}=Conv_{local\_embed\_Ni}\left(Pi\right)*F_{ing\_act\_Ni}+F_{global\_embed\_Ni}$$(14)Ni=RepBlockFatt_fuse_Ni

### 2.2. Introduction of RepViTBlock

In the GD mechanism, the C2f module serves as the core of multi-scale feature fusion, integrating feature maps of different resolutions to enhance target representation. However, the traditional C2f module, which relies on a standard convolutional structure, faces limitations when applied to complex road defect detection tasks. Due to its fixed computational mode, standard convolution restricts the feature extraction capability, making it challenging to effectively capture fine-grained details of small targets, such as road cracks. Additionally, its rigid computational structure limits the potential for optimizing inference efficiency, making it difficult for the model to maintain high detection accuracy while minimizing computational overhead.

To address these issues, this paper introduces RepViTBlock [25] to enhance C2f, thereby constructing a new Rep-GD module (as shown in Figure 3). Compared to other lightweight models such as MobileNet and ShuffleNet, the Rep-GD module achieves a superior balance between computational efficiency and feature extraction capability. While MobileNet and ShuffleNet are designed with a strong emphasis on lightweight architecture, they often sacrifice feature representation, which is particularly limiting in complex tasks such as road defect detection. In contrast, the Rep-GD module, optimized with RepViTBlock, ensures high detection accuracy while significantly reducing computational cost during inference.

RepViTBlock incorporates the structural reparameterization (SR) strategy, employing a multi-branch structure during training to enhance feature learning. During inference, these multiple branches are fused into a single path, effectively reducing computational complexity while maintaining strong feature expressiveness. This optimization improves detection efficiency, making the model more suitable for real-time applications [26]. Additionally, to further enhance feature fusion, RepViTBlock integrates a lightweight token mixer and channel mixer. The token mixer, utilizing a 3 × 3 depth-wise convolution, extracts local spatial features to enhance the detection of small targets, while the channel mixer, leveraging a 1 × 1 convolution, facilitates inter-channel information interaction to improve global feature representation. This design not only preserves detection accuracy but also prevents a significant increase in parameter count, allowing the model to achieve stronger feature extraction while maintaining high computational efficiency.

Furthermore, RepViTBlock incorporates the squeeze-and-excitation (SE) attention mechanism [27], which dynamically adjusts the weights of different feature channels to enhance the model’s focus on critical features, such as crack edges. By selectively emphasizing key features, this mechanism effectively strengthens the model’s feature extraction capability, leading to improved accuracy in road defect detection. With this design, RepViTBlock achieves higher feature expressiveness and lower computational complexity, and demonstrates excellent performance in multiple vision tasks, especially in scenarios such as real-time target detection.

### 2.3. Wise-IoU Loss Function

In the YOLOv8 model, CIoU-Loss [28] is usually used as the bounding box regression loss function. CIoU-Loss takes into account the overlap area, center point distance, and aspect ratio consistency of the predicted box and the real box, which has a certain improvement on the regression accuracy of the target box. However, in the road defect detection task, there can be the following problems: small targets such as road cracks, which are easily caused by insufficient gradient due to the low quality of anchor frames and can affect the detection accuracy; low quality anchor frames in the complex pavement environment, which can cause more interference and reduce the generalization ability of the model; and CIoU-loss, which has insufficient gradient allocation for centroid offset and can lead to lower regression efficiency.

In order to solve these problems, this paper adopts Wise-IoU v1 [29] instead of CIoU-loss. A key feature of Wise-IoU v1 is its dynamic non-monotonic focusing mechanism. Traditional focusing mechanisms are usually static, using fixed strategies to adjust the sample weights. However, these strategies may have limited effectiveness when dealing with low-quality anchor frames. In contrast, the dynamic non-monotonic focusing mechanism dynamically adjusts the gradient contribution of anchor frames based on their overlap (i.e., IoU) with target frames. This mechanism effectively mitigates the negative impact of low-quality anchor frames on model training, while enhancing the contribution of normal-quality anchor frames, allowing the model to more accurately learn valid regression information.

Wise-IoU v1 also introduces offset penalties and a weighting factor, which improves the accuracy of anchor frame quality assessment and effectively reduces the interference of low-quality anchor frames on gradient computation. By introducing the weighting factor $R_{WIoU}$, the weights of anchor frames are dynamically adjusted based on their quality, reducing the impact of low-quality anchor frames on model training, thereby improving the model’s robustness in complex environments. This weighting factor effectively mitigates the gradient interference from low-quality anchor frames and ensures the accurate flow of gradients, avoiding the regression inefficiencies and insufficient gradients that can occur in traditional CIoU-loss. Furthermore, Wise-IoU v1 enhances regression accuracy by precisely matching the centroids of the predicted and ground truth frames. The formula for Wise-IoU v1 is(15)$$L_{WIoUv1}=R_{WIoU}L_{IoU},L_{IoU}=1-IoU$$(16)$$R_{WIoU}=\exp\left(\frac{{\left(x-x_{gt}\right)}^{2}+{\left(y-y_{gt}\right)}^{2}}{\left(W_{g}^{2}+H_{g}^{2}\right)*}\right)$$
where IoU is the traditional intersection and fusion over union, $R_{WIoU}$ is a weighting factor used to adjust the weights of anchor frames of different quality, $\left(x,y\right)$ is the centroid coordinates of the prediction frame, $\left(x_{gt},y_{gt}\right)$ is the centroid coordinates of the true frame, $W_{g},H_{g}$ is the size of the minimum closed frame, and the symbol ∗ indicates that $W_{g},H_{g}$ is separated from the computational graph to effectively prevent them from interfering with the computation of the gradient and to avoid affecting the convergence process of the model.

## 3. Experiments and Results

In order to adapt to the characteristics of urban roads in China, this paper selects the Chinese regional road damage data in the RDD-2022 (Road Damage Dataset 2022) [30] dataset for model training. This dataset contains two parts: the China_Drone image set and the china_motorcycle image set. The China_Drone image set contains a total of 2401 road damage images, covering 3068 instances of road damage, while the china_motorcycle image set contains 1977 road damage images, covering 4650 instances of damage. All images are labeled with four types of road defects, including longitudinal cracks (D00), transverse cracks (D10), composite cracks (D20), and potholes (D40). Figure 4 shows four typical damage types in RDD-2022. To ensure data quality, some unlabeled road images were excluded from this study. The processed dataset was then randomly split into a training set and a validation set at a 7:3 ratio, with the training set containing 5530 images and the validation set containing 2370 images. Additionally, an independent test set consisting of 500 images, which was not involved in training or validation, was separately allocated to ensure the fairness and objectivity of the evaluation.

### 3.1. Evaluation Criteria

In this paper, the evaluation criteria of the model mainly include mean average precision (mAP), number of parameters (parameters), and number of billion floating point operations per second (GFLOPs). These indicators can comprehensively reflect the detection accuracy and computational efficiency of the model. The specific calculation formula is as follows:

Precision (precision, P) indicates the proportion of correctly detected targets, which is calculated by the following formula:(17)$$P=\frac{TP}{TP+FP}$$
where TP is the number of bit-positive samples correctly predicted by the model, and FP is the number of samples incorrectly predicted by the model.

Recall (recall, R), on the other hand, indicates the proportion of actual targets in the dataset that are detected and is calculated as(18)$$R=\frac{TP}{TP+FN}$$
where FN represents the number of undetected targets among the correct targets.

Average precision (AP) is the result obtained by integrating the precision-recall curve and is used as a measure of the model’s precision at different recall rates, which is calculated as(19)$$AP=\int_{0}^{1}P\left(R\right)dR$$
where P(R) is based on the recall rate, and R is the calculated precision rate.

In order to evaluate the performance of the model on different categories, this paper calculates the mean average precision (mAP). mAP is the average of the *AP* values of all categories, which is calculated as(20)$$mAP=\frac{1}{n}\sum_{i=1}^{n}AP\left(i\right)$$
where n denotes the number of categories, and AP(i) denotes the average precision of the ith category.

In this paper, mAP50 is chosen as an evaluation metric, which represents the mean Average precision (mAP) of all categories at an intersection over union (IoU) threshold of 0.5. IoU measures the degree of overlap between the predicted bounding box and the ground-truth bounding box, and it is calculated using the following formula:(21)$$IoU=\frac{\left|P\cap G\right|}{\left|P\cup G\right|}$$
where P is the area of the predicted bounding box, and G is the area of the labeled bounding box. With these metrics, this paper is able to comprehensively evaluate the detection accuracy, model complexity, and inference efficiency of the proposed model.

### 3.2. Ablation Experiments

In this study, ablation experiments are performed to verify the effectiveness of the proposed improved method. The GD mechanism, Rep-GD module, and the Wise-IoU loss function are added to the YOLOv8 model, respectively, and the experimental results of different module combinations are given. The experimental results are given in Table 1.

As shown in the table, the GD mechanism improves the ability of the model to detect targets of different sizes by improving the fusion of multi-scale features. The experimental results show that the addition of the GD mechanism improves the mAP50 by 0.8% and also improves the accuracy. Next, we introduced the RepViTBlock module on top of the GD mechanism, an improvement that further enhances the global feature extraction capability of the model by combining the properties of the C2f structure and the RepViTBlock. The experimental results show that compared to the model using only the GD mechanism, the improved GD mechanism (i.e., Rep-GD) improves the mAP50 by 0.7% while reducing the number of parameters by 6%. To further improve the model’s small target detection performance, we introduce the Wise-IoU loss function. This loss function improves the model’s sensitivity to edges and fine targets by optimizing the IoU calculation. Experimental results show that the addition of the Wise-IoU loss function improves the mAP50 by 1.4%.

The experimental results show that the combination of the three modules leads to a significant improvement in the overall performance of the model. Compared to the original YOLOv8 model, the improved model improves the mAP50 by 2.4% and the accuracy by 3%.

### 3.3. Comparative Experiments

In order to verify the performance advantages of the improved RepGD-YOLOV8W model in road defect detection, this paper presents a detailed comparison with several mainstream target detection algorithms. The comparison includes the traditional two-stage algorithm Faster R-CNN [31], the single-stage algorithm SSD [32], and the lightweight YOLO series algorithms, such as YOLOv5n, YOLOv5s, YOLOv7-tiny [33], and YOLOv8n, along with the improved ML-YOLO based on YOLOv8 and CA-YOLOv8 models. To ensure fairness and comparability, all experiments were conducted under the same hardware conditions and trained using a unified road defect detection dataset.

As shown in Table 2, RepGD-YOLOV8W significantly outperforms other algorithms in terms of detection accuracy. Compared to the traditional two-stage algorithm Faster R-CNN and the single-stage algorithm SSD, RepGD-YOLOV8W shows improvements of 9.1 and 9.5 percentage points, respectively. Among the YOLO series algorithms, RepGD-YOLOV8W also exhibits significant accuracy advantages, with mAP50 values that are 4.7, 3.9, 3.1, and 2.8 percentage points higher than YOLOv5n, YOLOv5s, YOLOv7-tiny, and YOLOv8n, respectively, demonstrating its powerful detection capabilities in complex road damage scenarios.

Faster R-CNN, as a classical two-stage target detection method, generates candidate frames through the region proposal network (RPN) and then classifies and regresses each candidate. Despite its superior accuracy, it suffers from slow detection speed and poor real-time performance, which makes it unsuitable for tasks requiring efficient, real-time detection, such as road disease detection. Furthermore, Faster R-CNN struggles with small targets and complex backgrounds, particularly when detecting small road defects like cracks, which often leads to missed or false detections. Thus, while accurate, Faster R-CNN is not practical for real-world applications.

SSD is a single-stage target detection algorithm widely used in various applications due to its high speed and real-time performance. SSD can handle targets of different sizes by detecting them across multiple feature maps. However, in road disease detection, SSD faces difficulties in accurately detecting small defects due to the low contrast of small targets, such as cracks, and interference from complex backgrounds. Additionally, the poor quality of bounding boxes generated by SSD is affected by low-quality anchor frames, which ultimately reduces detection accuracy.

Among the YOLO series algorithms, YOLOv5n and YOLOv5s perform well, especially YOLOv5n, which is lightweight and suitable for real-time detection tasks. However, the accuracy of detecting small targets in complex backgrounds is not as high as RepGD-YOLOV8W. YOLOv7-tiny offers higher accuracy, but compared to RepGD-YOLOV8W, it has slightly lower computational efficiency and higher model complexity, which affects its performance in resource-constrained environments. YOLOv8n is smaller than YOLOv5n in terms of model size and parameter count, but RepGD-YOLOV8W still has obvious advantages in accuracy and efficiency, especially in complex road damage scenarios.

In the comparison between ML-YOLO [34] and CA-YOLOv8 [35], although ML-YOLO improves accuracy by adopting the convolutional block attention mechanism for better feature extraction, its larger model size and computational volume result in slower inference speed. On the other hand, CA-YOLOv8 achieves some improvements in channel feature convolution and the C2f module, but its accuracy and frame rate still do not reach the level of RepGD-YOLOV8W, especially in handling small targets and complex backgrounds.

In terms of model parameters, RepGD-YOLOV8W has 5.80 M parameters, slightly higher than YOLOv5n (1.77 M) and YOLOv8n (3.01 M), but the improvement in accuracy is significant, indicating a good balance between performance and model size. In contrast, CA-YOLOv8 and ML-YOLO have larger parameter counts and fail to match the accuracy of RepGD-YOLOV8W.

Although RepGD-YOLOV8W performed well in several comparative experiments, there are still potential limitations to consider. In extreme road defect scenarios, such as heavily shaded or high-noise environments, the model’s performance may be challenged. Although the experimental results are based on the RDD2022 dataset, the coverage of this dataset is limited, failing to encompass all possible road defect types and environmental variables. Therefore, the model’s performance in other datasets or real-world scenarios still needs further validation. Future research should extend the dataset to improve the generalizability and stability of RepGD-YOLOV8W, ensuring its performance across a wider range of application scenarios.

In summary, the RepGD-YOLOV8W model significantly outperforms mainstream target detection algorithms, demonstrating its efficiency and practicality for road defect detection tasks.

### 3.4. Comparison of Test Results

The quantitative analysis results of RepGD-YOLOV8W and YOLOV8n in terms of mAP@50, precision, recall, and precision–recall (PR) curve are shown in Figure 5, Figure 6, Figure 7, Figure 8, Figure 9, Figure 10 and Figure 11. The experimental results show that YOLOv8n converges faster in the early stage of training. However, as training progresses, the mAP@50 of RepGD-YOLOV8W gradually surpasses that of YOLOv8n, maintaining the lead in the later stages, and ultimately achieving a slightly higher mAP@50, indicating superior overall detection accuracy. In terms of precision, YOLOv8n slightly outperforms RepGD-YOLOV8W in the early stage of training but with increasing epochs, and the precision of RepGD-YOLOV8W gradually improves, surpassing YOLOv8n in the later stages and eventually stabilizing. This suggests that RepGD-YOLOV8W has a lower false positive rate, leading to more accurate detection results. Regarding recall, YOLOv8n also dominates in the early stage, with a faster rise in recall rate. However, RepGD-YOLOV8W gradually improves in the later stages and ultimately surpasses YOLOv8n. This demonstrates that RepGD-YOLOV8W can detect more targets, enhancing the comprehensiveness of target recognition. Through the analysis of the precision–recall (PR) curve, RepGD-YOLOV8W outperforms YOLOv8n in overall mAP, achieving an mAP of 0.849 compared to 0.825 for YOLOv8n. In the detection of longitudinal cracks (D00), transverse cracks (D10), composite cracks (D20), and potholes (D40), RepGD-YOLOV8W shows improvements of 1.4%, 0.4%, 3.1%, and 4.3%, respectively. RepGD-YOLOV8W shows more significant improvements in the D20 and D40 categories, mainly because these targets have relatively larger and more distinct morphological features. This model can better utilize multi-scale feature extraction and global information fusion when detecting these larger targets, especially in complex backgrounds, where such targets are easier to detect compared to smaller cracks or fine defects. For smaller targets like D10, although the model shows improvements, the gain is relatively small due to the small size of the features and the complexity of background interference.

In order to further verify the performance advantage of the improved RepGD-YOLOV8W model in road defect detection, this paper conducted a comparison test with the original YOLOv8 model to analyze the detection effect of the two in different scenes. In the scenes shown in the first group, complex background interference, such as weeds, obviously increases the detection difficulty, and the original YOLOv8 model has leakage detection, while the improved REPGD-YOLOV8W model, which enhances the multi-scale feature extraction capability by introducing the Rep-GD module, is able to accurately detect all crack lesions. RepGD-YOLOV8W model is capable of accurately labeling highway diseases. In the second set of test images, the scene contains larger crack targets, but the original YOLOv8 model fails to detect these lesions completely, and there is some leakage. This may be due to insufficient depth of the feature representation to capture the global features of large targets. In contrast, the improved RepGD-YOLOV8W model is able to extract the crack feature information more comprehensively through the feature enhancement effect of RepViTBlock and successfully detect all cracks. In the third group of images, the original YOLOv8 model fails to detect the composite crack next to the manhole cover in the upper right corner due to the blurred defect area, while the improved algorithm is able to capture the key features of the fuzzy target by relying on its optimized feature extraction mechanism. In the fourth group, it can also be seen that in complex environments, the improved algorithm model performs more robustly compared to the original model, which not only reduces the leakage detection but also improves the accuracy and confidence of detection. The experimental results fully demonstrate the practical application potential and superior performance of the improved model in road defect detection tasks.

## 4. Conclusions

In this paper, a RepGD-YOLOV8W model based on YOLOv8 improvement is proposed to solve the problems of leakage and misdetection in road defects. The model improves the expression ability of multi-scale features by introducing the RepViTBlock, while effectively optimizing the inference efficiency. By using the Rep-GD mechanism, the feature interaction and fusion methods are improved, which enhances the detection performance of the model for targets of different scales. In addition, the introduction of the Wise-IoU loss function makes the model more accurate in detecting both small and large targets. The experimental results show that RepGD-YOLOV8W achieves even better results than the YOLOv8n model on the RDD2022 dataset, with a 2.4% improvement in mAP50, and is able to meet real-time detection requirements. The model maintains a better balance between computational efficiency and accuracy of road defect detection and has stronger practicality. Future research will focus on further optimizing the algorithm to improve the accuracy of the model without increasing the computational resources and exploring its feasibility in a wider range of application scenarios.

## Figures and Tables

**Figure 1 sensors-25-01287-f001:**
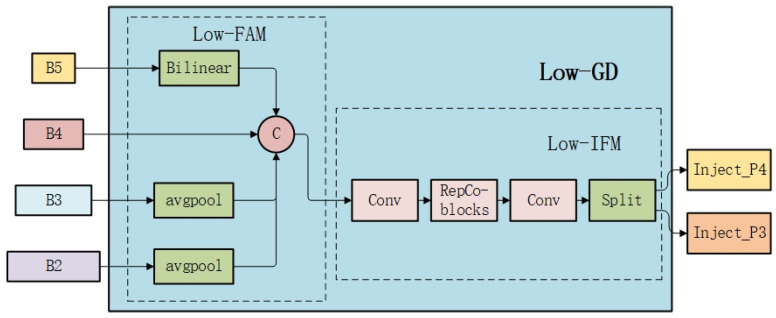
Low-stage collection-distribution branches.

**Figure 2 sensors-25-01287-f002:**
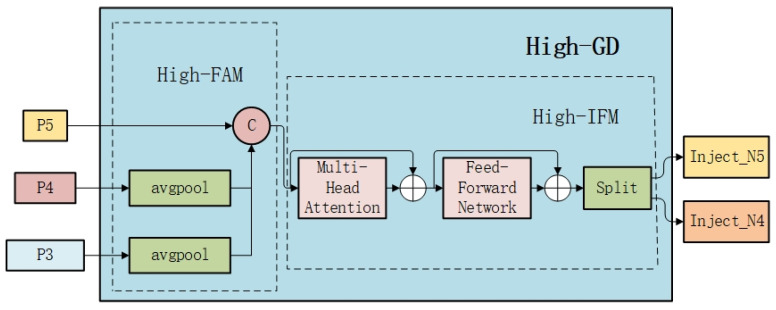
High-stage collection-distribution branches.

**Figure 3 sensors-25-01287-f003:**
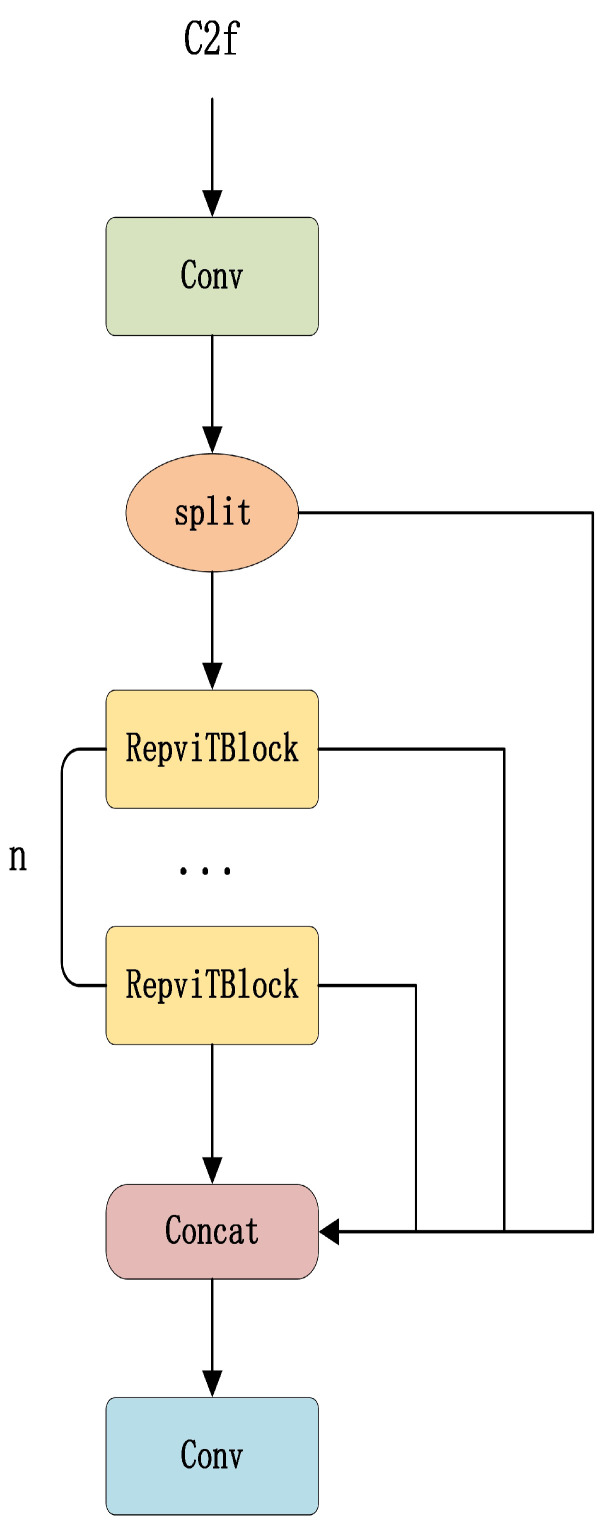
Improved C2f module diagram.

**Figure 4 sensors-25-01287-f004:**
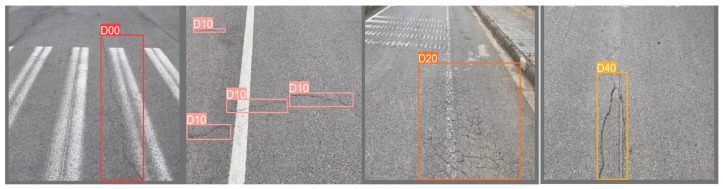
Four typical damage types.

**Figure 5 sensors-25-01287-f005:**
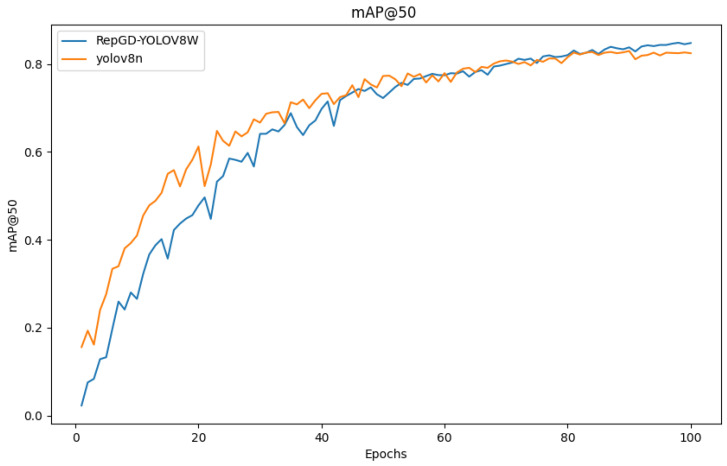
mAP@50 comparison chart.

**Figure 6 sensors-25-01287-f006:**
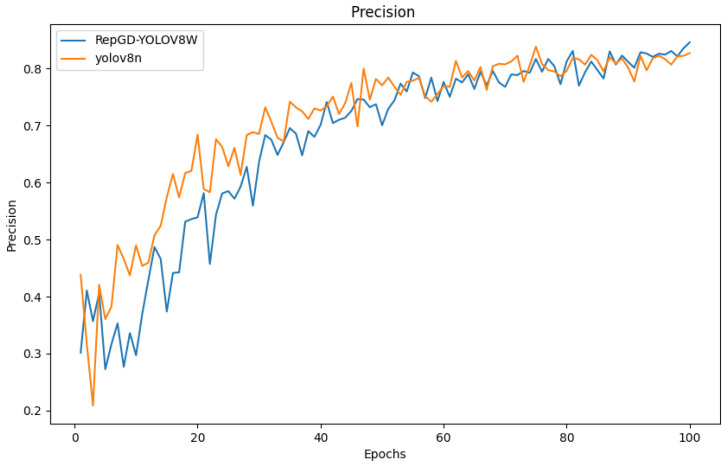
Precision comparison chart.

**Figure 7 sensors-25-01287-f007:**
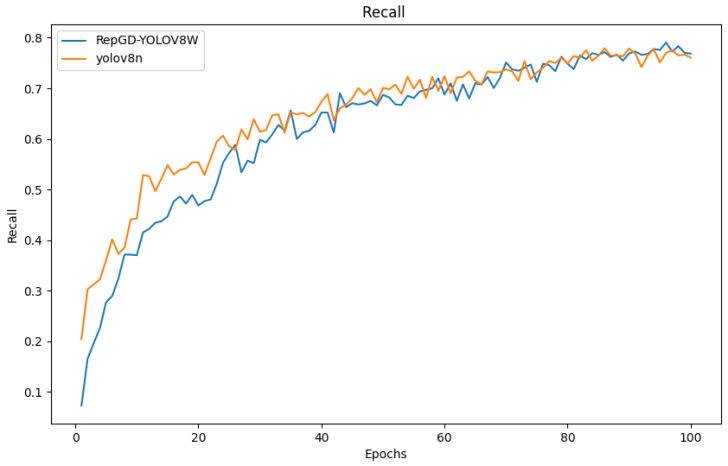
Recall comparison chart.

**Figure 8 sensors-25-01287-f008:**
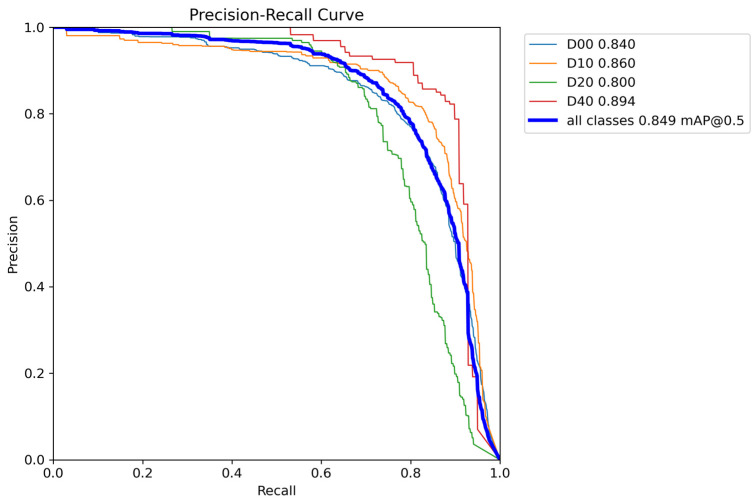
RepGD-YOLOV8W PR curves.

**Figure 9 sensors-25-01287-f009:**
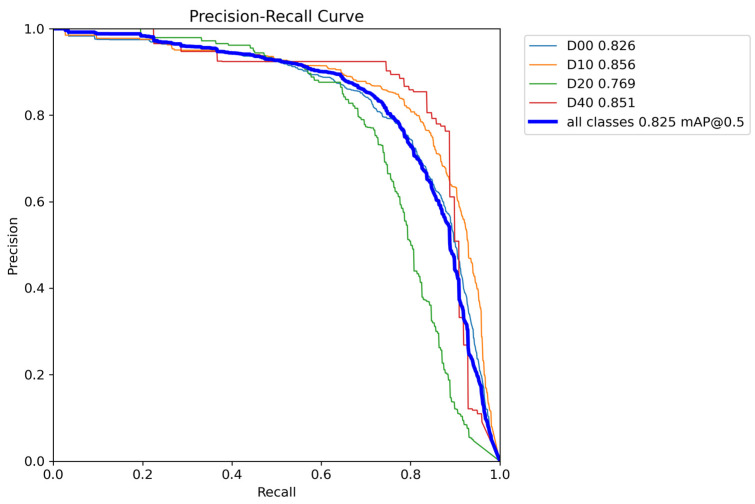
YOLOv8n PR curves.

**Figure 10 sensors-25-01287-f010:**
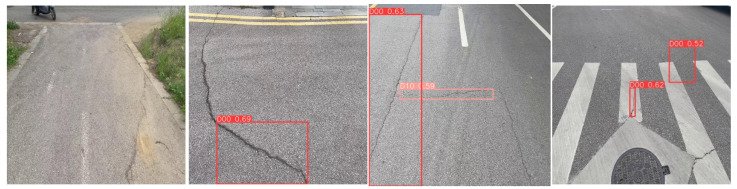
YOLOv8 detection results.

**Figure 11 sensors-25-01287-f011:**
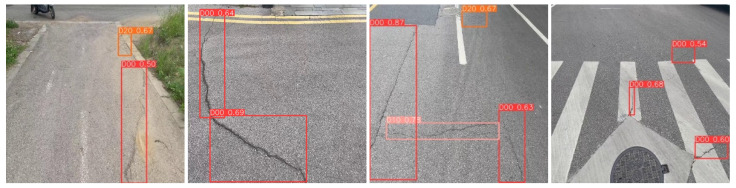
RepGD-YOLOV8W test results.

**Table 1 sensors-25-01287-t001:** Results of ablation experiments.

Model	P%	R%	map50%	Params	GFLOPs
YOLOv8n	0.939	0.95	0.825	3,230,908	8.4
YOLOv8n+GD	0.947	0.95	0.833	6,227,228	12.3
YOLOv8n+Rep-GD	0.941	0.94	0.840	5,822,236	11.7
YOLOv8n+Wise-IoU	0.946	0.94	0.839	3,230,908	8.4
YOLOv8n+GD+Wise-IoU	0.904	0.94	0.836	6,227,228	12.3
YOLOv8n+Rep-GD+Wise-IoU	0.969	0.94	0.849	5,822,236	11.7

**Table 2 sensors-25-01287-t002:** Comparative experimental results.

Model	mAP50%	Params/M
Faster R-CNN	0.758	41.14
SSD	0.754	24.83
YOLOv5n	0.802	1.77
YOLOv5s	0.810	7.03
YOLOv7tiny	0.818	6.02
YOLOv8n	0.825	3.01
ML-YOLO	0.840	139.5
CA-YOLOv8	0.839	7.40
RepGD-YOLOV8W	0.849	5.80

## Data Availability

Data are contained within the article.

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
