# Peer review of "Deep Learning-Based Algorithm for Road Defect Detection"

_sensors, 2025, doi:10.3390/s25051287_

Round 1
Reviewer 1 Report
Comments and Suggestions for Authors Comments on the Quality of English LanguageThe paper contains numerous instances where the writing is unclear and imprecise, making it difficult for the reader to understand the authors' intended meaning. For example, in the Introduction section, the paper mentions "challenges remain, such as insufficient detection precision for small targets and issues of missed or false detections in complex backgrounds," but it fails to provide a clear and concise explanation of why these challenges are significant or how they affect the field of road defect detection. This lack of clarity undermines the paper's overall readability and comprehension.
Reviewer 2 Report
Comments and Suggestions for Authors
The authors have proposed a deep learning-based algorithm for road-defect detection. Overall, the work is good. However, the paper needs thorough English correction and proof reading.
A few suggestions to the authors:
1) All the acronyms must be expanded before they’re first used in the text.
2) Nothing is clear from Figures 1 and 2.
3) More Figures should be provided to show the efficiency of the proposed algorithm.
4) Performance of the algorithm should be compared with the state-of-the-art methods.
Comments on the Quality of English LanguageThe authors have proposed a deep learning-based algorithm for road-defect detection. Overall, the work is good. However, the paper needs thorough English correction and proof reading.
A few suggestions to the authors:
1) All the acronyms must be expanded before they’re first used in the text.
2) Nothing is clear from Figures 1 and 2.
3) More Figures should be provided to show the efficiency of the proposed algorithm.
4) Performance of the algorithm should be compared with the state-of-the-art methods.
Reviewer 3 Report
Comments and Suggestions for Authors
Comments to the Author(s)
This manuscript presents
Deep Learning Based Algorithm for Road Defect Detection. This paper contains a good effort related to the monitoring of highway infrastructures. The topic is original. The methodology of the study is well explained. In general, the manuscript is organized well. The study offers reliable findings and is supported by sufficient proof. It could be accepted for publication if the authors resolve the following issues. All the answers should be included in the manuscript.
1. Please put all the reference numbers in quotation signs (e.g., [1]).
2. Add at least 10 recent relevant references to the introduction.
3. Add a Figure to show the common highway diseases from the used RDD2022 dataset.
4. Add more details to the captions of Fig.1 and Fig. 2.
5. Add some references to the equations of section 2.1.1 and 2.1.2.
6. Section 3, the data was divided into a training set and validation set according to the ratio of 7:3. How did you test the system without a test dataset?
7. Figure 5. Can the RepGD-YOLOV8W label of highway diseases? Please clarify that.
Round 2
Reviewer 1 Report
Comments and Suggestions for Authors
- The paper mentions that the processed dataset is divided into training, validation, and test sets. However, it would be helpful to explicitly state the exact number of images in each set or the proportion used for validation and testing. This information is crucial for reproducibility and understanding the robustness of the model evaluation.
- While the paper includes visual results of the model's performance, it would be beneficial to add more detailed visualizations, such as precision-recall curves or confusion matrices, to provide a more comprehensive view of the model's performance across different classes of road defects. This would help readers better understand the model's strengths and areas for improvement.
